# Disparities in Healthcare Utilization: Superfund Site vs. Neighboring Comparison Site

**DOI:** 10.3390/ijerph19159271

**Published:** 2022-07-28

**Authors:** Crystal Stephens, Young-il Kim, Rekha Ramachandran, Monica L. Baskin, Veena Antony, Sejong Bae

**Affiliations:** 1Division of Pulmonary, Allergy and Critical Care Medicine, School of Medicine, University of Alabama at Birmingham, Birmingham, AL 35294, USA; ctstephens@uabmc.edu (C.S.); vantony@uabmc.edu (V.A.); 2Division of Preventive Medicine, School of Medicine, University of Alabama at Birmingham, Birmingham, AL 35294, USA; youngkim@uabmc.edu (Y.-i.K.); rramachandran@uabmc.edu (R.R.); mbaskin@uabmc.edu (M.L.B.)

**Keywords:** pollution, Superfund, health disparities

## Abstract

Inequities in pollution-attributable health disparities are similar in most urban areas throughout the United States, and appear to encompass racial and socio-demographic differences, thereby suggesting increased health risks for those living in these areas. Individuals residing in close proximity to Superfund sites, predominantly people of color, are increasingly stricken with lung diseases. The prevalence of chronic lung diseases, such as chronic obstructive pulmonary disease (COPD), asthma in children, and lower respiratory tract infections (LRTI), is significantly higher in the affected area compared to the neighboring control area, irrespective of smoking, socio-economic status, or demographics. We conducted a retrospective analysis using data collected from patients who obtained healthcare from the University of Alabama at Birmingham (UAB) Health System. The data were procured from the Enterprise Data Warehouse (UAB Informatics for Integrating Biology and the Bedside (i2b2)). We evaluated healthcare utilization and classification of disease (defined by ICD-10 codes) of patients residing in zip codes: affected (35207, 35217) and neighboring comparison (35214). The results of the analysis may provide evidence that can be used for risk mitigation strategies or outreach education campaign(s) for those who live in the affected area.

## 1. Introduction

Coal is the world’s most polluting fossil fuel, and coal combustion is an important cause of climate change [1]. High-temperature processes, such as smelting and coal combustion, are typically associated with the generation of fine particulate (≤2.5 μm), termed particulate matter or PM_2.5_ [2]. Particles this small are easily inhaled and deposited in the lower airway, where they can cause irritation, chronic inflammation, and/or exacerbation of pre-existing lung conditions. Jefferson County, Alabama, is located in the north central region of the state and it is the center of the former iron, coal, and limestone mining belt of the southern United States [3]. Sitting at the center of Jefferson County is the city of Birmingham, an urban, metropolitan area. There lies, within this area, multiple locations of actively functioning coke furnaces [3]. 

In 2011, after the United States Environmental Protection Agency (EPA) discovered toxic contaminants leaching offsite from one of the local coke plants, the EPA utilized its emergency Superfund authority to become involved in this North Birmingham neighborhood [4]. The Superfund program allows the EPA to identify and hold liable potentially responsible parties (i.e., generators or transporters of the hazardous waste on the site, and past or present owners of the site), and establishes a trust fund for site clean-up, when no responsible party is identified [5]. The 35th Avenue Superfund site in North Birmingham is located in the affected zip code 35207, with adjacent zip code 35217 [6]. The efforts surrounding remediation and clean-up of the Superfund site are ongoing, and although responsible parties have been identified, adding the location to the National Priorities List (NPL) has not yet occurred [7]. In addition to the Superfund site, the EPA’s Toxics Release Inventory (TRI) in 2019 reported that the North Birmingham area contains the top four industries responsible for chemical releases [4]. 

Although external factors within the environment, such as air, soil, and water pollution, are well recognized for having negative health impacts on the population, the results of our data analysis have identified other factors (e.g., differences in location leading to the delineation of vicinity, for which residents experience increased prevalence of disease) that draw our attention to how these situations have been resolved in other areas experiencing similar conditions. Environmental exposures are complex, both geochemically and geographically [8]. Concrete research linking the environment to human health is beginning to mature, highlighting the subtlety of multiple exposures and mixtures of substances, along with the cumulative legacy effects of life in disrupted and stressed environments [8]. 

Pollution is the largest environmental cause of disease and premature death in the world today [1]. Pollution and its related diseases largely affect the world′s poor and powerless [9]. Diseases caused by pollution were responsible for an estimated nine million premature deaths in 2015. That is, 16% of all deaths worldwide, three times more deaths than from AIDS, tuberculosis, and malaria combined, and fifteen times more deaths than from all wars and other forms of violence [10]. Pollution disproportionately kills the poor and the vulnerable. Nearly 92% of its related deaths occur in low- and middle-income countries and in other countries at every income level; disease caused by pollution is most prevalent among minorities and the marginalized [1].

Although some mechanistic details remain incompletely described, the existing science is deemed adequate to substantiate several plausible biological pathways whereby PM_2.5_ could instigate acute cardiovascular events and promote chronic disease [11]. Fine particulate matter (i.e., PM_2.5_) is the best studied form of air pollution and it is linked to a wide range of diseases in several organ systems [1]. The strongest causal associations are seen between PM_2.5_ pollution and cardiovascular and pulmonary diseases. Specific causal associations have been established between PM_2.5_ pollution and myocardial infarction, hypertension, congestive heart failure, arrhythmias, and cardiovascular mortality [1]. Causal associations have been established between PM_2.5_ pollution and chronic obstructive pulmonary disease (COPD) and lung cancer [1]. The International Agency for Research on Cancer (IARC) has reported that airborne particulate matter and ambient air pollution are proven group 1 human carcinogens [12]. 

A slowly progressing disease, characterized by the gradual loss of lung function, COPD causes 4% of overall global disease burden [10]. The most important COPD risk factor is active smoking, but other risk factors are occupation (e.g., coal and hardrock mining), environment, and socio-economic deprivation in childhood [13]. Asthma, an inflammatory respiratory condition, is a major cause of disability, healthcare utilization, and reduced quality of life, and it accounts for approximately 1% of overall global disease burden [10]. Inhaled PM_2.5_ induces oxidative stress, leading to inflammatory airway responses and bronchial hyper-reactivity; metal fumes are well-recognized causes of occupational asthma, but the contribution of metal dust to non-occupational asthma is not as straightforward [8].

### 1.1. Jefferson County, Alabama

Jefferson County, Alabama, consists of 55 zip codes, including the affected areas of 35207 and 35217. The population of Jefferson County exceeds 600,000 residents with a racial profile of 52.4% White, 42.5% Black, and other, as obtained from the U.S. Census Bureau [14]. The Jefferson County Community Health Equity Report, published statistics on disability status (15–32.6% affected vs. 20–24.9% control), poverty level (28.4–64.5% affected vs. 16.1–28.3% control), life expectancy (61.8–70.1 years affected vs. 70.2–73.7 years control), and infant mortality (18.4–34.1% affected vs. 10.2–18.3% control), between census tracks for the residents in zip codes 35207 and 35217, as compared to neighboring zip code 35214, as well as racial and ethnic distribution and healthy food access showing higher concentrations of racial and ethnic minorities in the areas of food deserts [3]. 

Inequities in pollution-attributable health disparities are similar in most urban areas throughout the United States and appear to encompass racial and socio-demographic differences, thereby suggesting increased health risks for those living in these areas (i.e., predominantly people of color). These health outcomes have been determined to be causally associated with PM_2.5_ by varying sources, including the U.S. EPA [15] and the Global Burden of Disease (GBD) study [16]. 

### 1.2. Community Impact

Lower income, minority, and marginalized populations experience higher air pollution exposure levels and associated health impacts [9,17]. The communities surrounding the 35th Avenue Superfund site are directly exposed and, thus, vulnerable to major air pollution sources. This burden on population health, born decades ago out of industrial enterprise, continue to disproportionately affect these communities, posing a public health threat associated with negative health outcomes from long-term air pollution exposure [18]. Understanding the various measures of self-reported poor health and the accompanying outcomes experienced by the community is critical for timely and effective public health intervention. 

## 2. Materials and Methods

To accelerate translational biomedical research for the Birmingham, Alabama Superfund site, we explored UAB Informatics for Integrating Biology and the Bedside (i2b2), a database of electronic health records (EHRs) with the co-occurrence of frequencies for conditions and demographics (e.g., sex, race, and ethnicity), marital status observed for UAB Health System patients; specifically 817,794 records from inpatient (IP), outpatient (OP), and emergency room (ER) visits (dichotomous) from 27,205 unique patients for this particular study from 2011 to 2021. This patient number accounts for 64.3% of the total population in zip codes 35214, 35207, and 35217. The 35th Avenue Superfund site is located in North Birmingham and has industries such as coke oven plants, steel producing facilities, and a recycling plant, among others [6]. The affected area, consisting of zip codes 35207 and 35217, was chosen due to the communities’ location within the Superfund site. An adjacent community with similar demographics and socio-economic status, but without industrial production, was selected as the control area, zip code 35214. 

Variables of interest, such as demographics, Elixhauser comorbidity indices, number of patients with IP visits, number of patients with OP visits, and number of patients with ER visits, were compared. These data were then analyzed using the International Classification of Diseases, 10th Revision (ICD-10) diagnosis codes. Data were restricted to the adult population (≥18 years of age) and to patients residing within zip codes: 35207 and 35217 for the affected group, and zip code 35214 for the neighboring group; all within 5 miles of the academic medical center located in the downtown area of Birmingham. The Chi-square test was used to compare zip code 35214 (neighboring area) and zip codes 35207 and 35217 (affected area). Logistic regression adjusted odds of any healthcare utilization by setting was used to compare the healthcare visits.

## 3. Results

### 3.1. Characteristics of the Population

The characteristics of the 15,144 patients (affected area) and 12,061 patients (neighboring control area) are shown in Table 1. The participant mean age was 50.96 (affected) and 51.89 (neighboring), and the majority of residents were black in both the affected and neighboring areas (affected—72% black, 18.4% white; neighboring—75.8% black, 17.8% white). However, the neighboring control were less likely to report a self-assessment health condition rating of poor, while the affected population was more than twice as likely to rate their health as poor (affected 12.2%; neighboring 5.6%) (Table 1).

After examining the association between zip code and increased chronic comorbid conditions, we considered additional socio-demographic factors, such as financial concern and education level. The results are consistent in that the affected population reported a higher level of financial concern (31.6%) as compared to the neighboring (23.6%) (Table 1). This information was collected through a standard questionnaire to assess patient social history at routine health evaluation or hospital admission. Similarly, compared to the neighboring group, the affected population had a lower percentage of education beyond high school level (34.8% affected, 51.2% neighboring), and also a higher percentage of individuals with a less than high school education (51% affected, 37.2% neighboring) (Table 1). Upon further assessment of group social history with information collected from EHR, there was a higher rate of current smoking and substance abuse among the affected (37.6%, 14%) versus the neighboring (31%, 10.8%), respectively, but lower current alcohol use in the affected population (36.6% affected, 40.4% neighboring) (Table 1). 

Patients living in the affected zip codes were less likely (75.9%) to utilize outpatient visits and were more likely to have inpatient visits (36%), emergency room visits (71.0%), and longer time in inpatient and emergency (Table 2). Overall, affected area patients were found to be 1.32 times and 1.31 times more likely to have IP and ER visits, respectively, compared to neighboring, and the affected patients were less likely to have OP visits, adjusted odds ratio (OR) (0.86: 95% confidence interval (CI) 0.80, 0.91) (Table 3).

Our data analysis reveals that the affected community has a higher report of alcohol abuse (affected 3.2%, neighboring 2.2%), drug abuse (affected 7.6%, neighboring 5%), psychoses/mental illness (affected 3.2%, neighboring 2%), and depression (affected 10.4%, neighboring 9.4%) (Table 4).

### 3.2. Overall Association between Zip Code and Increased Risk for Health Effects

The prevalence of chronic lung diseases, such as COPD, asthma in children, and lower respiratory tract infections (LRTI), is significantly higher in the affected area compared to the neighboring control area, irrespective of smoking, socio-economic status, or demographics. The data reveal supporting evidence that increased pollutant exposure may have affected numerous chronic conditions and hospital visits. In logistic regression models for hospital admission, adjusted for age, race, gender, and marital status, there was an association between diagnostic coding, with higher prevalence among participants from the affected area. Specifically, comparing disease conditions between the affected and neighboring zip codes, we observed higher chronic comorbidity prevalence in the affected area; relevant chronic conditions, such as congestive heart failure (6.4% affected, 5.6% neighboring), chronic pulmonary disease (12% affected, 9.8% neighboring), liver disease (4.6% affected, 4% neighboring), and other neurological conditions (7.2% affected, 6.4% neighboring) (Table 4). In addition, the affected are participants presented with higher alcohol abuse, drug abuse, psychoses (mental illness), and depression. 

## 4. Discussion

### Increased Chronic Disease

In this population-based comparison study, we found an association between residing in the affected zip code areas (35207 and 35217) and increased prevalence of multiple chronic diseases. In particular, pulmonary diseases (COPD), congestive heart failure, myocardial infarction, liver disease, and other neurological conditions were shown to be higher, even after adjusting for potential confounding factors (age, race, gender, and marital status). 

Colmer et al. [18] analyzed 36 years of data using geographic, economic, and demographic data from 65,000 U.S. census tracts and found, over the interval, that the spatial distribution of fine particulate matter concentrations has remained largely unchanged in the U.S. The result of this dataset analysis determined: (i) the most polluted census tracts in 1981 remained the most polluted in 2016; (ii) the least polluted census tracts in 1981 remained the least polluted in 2016; and (iii) the most exposed subpopulations in 1981 remained the most exposed in 2016. Although strides in pollution control have been made, we have been less successful in addressing exposure disparities between communities [18]. 

Pollution-attributable health risks and air exposure pollution levels are found to be higher in the affected area, resulting in a disparate distribution of negative health effects. The soil sample assessment focus for this area as requested by the EPA was of arsenic, lead, and polycyclic aromatic hydrocarbons (PAHs). The Agency for Toxic Substances and Disease Registry (ATSDR) concluded that past and current exposures to these substances could lead to harmful health outcomes, especially in children, where 5.2% of residential properties tested had arsenic levels of concern, as well as 11.8% of residential properties where soil was at or in excess of 1 per 10,000 people, the cancer threshold for arsenic [4]. Although the ATSDR further concluded that long-term exposure to PAHs increases the risk of cancer, the Jefferson County Department of Health’s geospatial analysis did not find significant differences between the North Birmingham zip codes and the rest of Jefferson County. These metrics were based on overall mortality and not the totality of chronic disease burden that reflects our data analysis [4]. In the setting of all else being similar, we report that an individual living in zip code 35207 or 35217 (affected group) has an increased chance of having a chronic disease compared with the matched counterpart in zip code 35214 (neighboring group). This association suggests that zip code or geographic location is an important consideration in the development and severity of disease and should direct future research to define the causes, precipitating factors, and barriers, thereby aiding in the proposal of changes to existing systems to better support the removal of environmental insults. These essential steps would lead to the potential reduction in disease burden and promote positive change for the affected population. 

Hospital utilization among the affected population did not tend to follow a manner of prevention or early intervention. Primary prevention (e.g., routine wellness visits) helps to promote health and well-being among populations [19]. Secondary prevention (e.g., screenings and other health examinations) provides patients with early intervention, which utilizes guidelines to better monitor and address concerns that may develop through the aging process. Furthermore, it enables the healthcare team to partner early on with patients to identify, manage, and treat conditions [19]. The affected population had an 0.86 times lower OP visit ratio compared with the neighboring population in our study, thus reducing the opportunity to evaluate symptomology, which may precipitate acute sickness that warrants escalated attention by the healthcare system. These findings suggest that affected residents are less likely to receive routine healthcare to better manage chronic conditions and, instead, utilize emergency care at higher levels, which may subsequently require IP hospitalization. 

Environmental injustice is the inequitable exposure of poor, minority, and disenfranchised populations to toxic chemicals, contaminated air and water, unsafe workplaces, and other forms of pollution. Consequently, these populations experience a disproportionate burden of pollution-related disease, often in violation of their human rights [20]. To advance environmental justice and reduce the inequitable exposure of the poor and marginalized, countries must develop legal mechanisms that provide recourse for environmental injustice [1]. Holding industries accountable if they are found to be operating outside of federal or state regulation guidelines is one way in which to support the injustice of overly polluted areas. More specifically for Jefferson County, we must explore the ways in which the environment affects health, and initiate strategies that positively impact choices, behaviors, and outcomes [3]. Further studies would need to be conducted to better understand the stressors that people living in disenfranchised communities experience, and associations with alcohol or drug use as coping mechanisms should be explored. In addition, more community engagement would be needed to identify the lived experiences of community members, and to quantify the ways in which living in environmentally unsafe areas affects their lives, and any behavioral influences that may result. 

Our study analyzed additional variables, which revealed an increase in drug and alcohol abuse and mental illnesses including depression. The poverty level difference reported earlier supports the common feature of lower-income neighborhoods, which are predominantly composed of minorities, being subject to environmental abuse, and they often have the least number of resources to address the issue or the industry. It is reported that 70% of hazardous waste sites officially listed on the National Priorities List (NPL) are located within one mile of federally assisted and subsidized housing [21]. Relocation would be the most complete resolve for these community members, but lower socio-economic status prohibits people from being able to relocate, and likely further compounds the stress and financial strain of providing for a family. We have mentioned the reports of food deserts and food scarcity and addressed poor access to health care. These issues continue to burden the community, and can aggravate external stress factors which may lead to chronic depression and subsequent abuse of substances. Though it is often impossible to determine (with certainty) that a given exposure caused a disease in an individual patient, many clinicians would not hesitate to link a significantly lengthy history of tobacco use or chain smoking to a patient’s lung cancer. Therefore, it may be reasonable to view certain environmental exposures in the same way [22], and the totality of the environmental insults and the social determinants of health on behavioral outcomes are worth future exploration. 

Strengths of this study include: (i) availability of and access to detailed datasets for our affected area of interest, enabling us to obtain specific patient-reported data with regard to socio-demographics, documented diagnoses, and associated comorbidities; (ii) utilization of a matched neighboring population with the equivalent data availability; and (iii) adjustment for potential confounders based on age, race, gender, and marital status. Limitations that bring about incomplete data interpretation include possible selection bias (given that our data collection only allows for documented medical encounters and does not encompass the entire population), as well as those residents seeking and utilizing healthcare services at nearby or surrounding facilities (i.e., does not capture a comprehensive representation of the area). Second, as investigators continue to identify social history aspects that may exacerbate certain conditions, we do not have a complete picture of whether these perceived barriers negatively amplified the health of the affected population more so than the neighboring population. Based on prior literature, we can therefore only postulate that certain factors, such as limited healthcare access, food scarcity, and increased environmental insults, can “fuel the flames” of poor health and wellness; we cannot conclude how these factors contribute to health outcome variation. Thirdly, the lack of data on residential history and subsequent length of exposure to environmental insults in our zip codes is unknown. Additionally, data on the outpatient visits is unknown in comparison to the health outcomes that we analyzed. Lastly, our findings may not be generalizable to all geographic areas, given the differences in social factors, environmental exposure, access to and utilization of healthcare facilities, and chronic disease burden experienced by our affected population. These limitations make it difficult to further refine the exact explanations for the dissimilarities in health status between our affected and control areas, based on social and environmental factors alone. 

## 5. Conclusions

Pollution is associated with race, poverty, and demographic factors, owing to socio-political forces, location of residence, and other determinants [18]. Continued efforts to mitigate air pollution and its detrimental effects on the Superfund community are vital to bring about positive change to area residents. These baseline disease numbers can be utilized to further investigate the negative impacts of increased air pollution and associated health burdens. Attempting to quantify exposure and its health impact is challenging, but there is enough evidence to recognize that there is a correlation that leads to substantial adverse health outcomes. As we continue to strive for progress in health promotion and disease prevention in society, the study investigators’ analysis may provide evidence that can be used for risk mitigation strategies or outreach education campaign(s) for those who live in the affected area. 

In conclusion, residents of the Superfund affected area of Jefferson County (zip codes 35207 and 35217) are experiencing disproportionate negative health outcomes, higher disease burden, and increased exposure to heavy metal emissions and other sources of air, soil, and water pollution. This community is owed a closer exploration of the correlations between their health and the impact of the environment in which they live and work. 

## Figures and Tables

**Table 1 ijerph-19-09271-t001:** Socio-demographics, behaviors, and health condition by neighboring (control) vs. Superfund site (affected).

	Neighboring Control Zipcode (35214)		Affected Zipcode (35207, 35217)		*p*-Value
Number of patients *	12,061	61.56	15,144	66.60	
Age, Mean (SD)	51.89 (19.16)		50.96 (19.05)		<0.0001
**Gender**					
Male	5244	43.4	6898	45.6	0.0007
Female	6814	56.6	8245	54.4	
**Race**					
White	2044	17.8	2689	18.4	<0.0001
Black	8690	75.8	10,506	72	
Other	729	6.4	1404	9.6	
**Marital Status**					
Married/Life Partner	3653	32.8	3583	25.2	<0.0001
Single	5704	51.2	8303	58.2	
Divorced/Separated/Widowed	1781	16	2379	16.6	
**Smoking Status**					
Current	1033	31	1578	37.6	<0.0001
Former	527	15.8	767	18.2	
Never	1777	53.2	1861	44.2	
**Substance Use**					
Current	765	10.8	1303	14	<0.0001
Former	289	4	530	5.8	
None	6092	85.2	7446	80.2	
**Alcohol Use**					
Current	3094	40.4	3609	36.6	<0.0001
Former	346	4.6	574	5.8	
None	4215	55	5658	57.4	
**Health Condition**					
Excellent/Fair	230	53.2	270	50.8	0.0016
Good	178	41.2	196	37	
Poor	24	5.6	65	12.2	
**Education**					
Less than high school	354	37.2	731	51	<0.0001
High school graduate/GED	111	11.6	205	14.2	
Greater than high school	489	51.2	498	34.8	
**Financial Concerns**					
Yes	140	23.6	284	31.6	0.0007
No	454	76.4	614	68.4	

* Number of unique patients in UAB i2b2 database from target zip code(s) and percentage of population in the zip code(s). Totals may not add to total due to missing data.

**Table 2 ijerph-19-09271-t002:** Health visit utilization.

Health Visit Utilization	Neighboring Control Zip Codes N %		Affected Zip Codes N %		*p*-Value
Number of patients with any Inpatient visits	3570	29.6	5454	36.0	<0.0001
Number of patients with any Outpatient visits	9556	79.2	11,498	75.9	<0.0001
Number of patients with any Emergency Room visits	7786	64.6	10,756	71.0	<0.0001
Average length of Inpatient visit, stay per patient, Mean (95% CI)	4.18 (3.90, 4.47)		5.02 (4.76, 5.29)		<0.0001
Average Inpatient visits per patient, Mean (95% CI)	0.71 (0.68, 0.75)		0.89 (0.85, 0.92)		<0.0001
Average Outpatient visits per patient, Mean (95% CI)	22.37 (21.57, 23.17)		21.75 (20.94, 22.56)		0.288
Average Emergency room visits per patient, Mean (95% CI)	2.27 (2.19, 2.35)		2.95 (2.86, 3.05)		<0.0001

CI: Confidence Interval.

**Table 3 ijerph-19-09271-t003:** Adjusted health visit utilization.

	Adjusted Odds Ratio	95% Confidence Interval		*p*-Value
Inpatient (IP) visit	1.32	1.25	1.39	<0.0001
Outpatient (OP) visit	0.86	0.80	0.91	<0.0001
Emergency Room (ER) visit	1.31	1.24	1.39	<0.0001

Adjusted for age, race, gender, and marital status.

**Table 4 ijerph-19-09271-t004:** Prevalence of Elixhauser comorbidity conditions stratified by location.

Disease Condition	Neighboring Control N %		Affected N %		*p*-Value
Congestive heart failure	667	5.6	978	6.4	0.0014
Cardiac arrhythmias	989	8.2	1399	9.2	0.0026
Valvular disease	323	2.6	413	2.8	0.8042
Pulmonary circulation disorders	299	2.4	386	2.6	0.7151
Peripheral vascular disorders	514	4.2	712	4.8	0.0823
Hypertension, uncomplicated	3384	28	4285	28.2	0.6651
Hypertension, complicated	790	6.6	1002	6.6	0.8263
Paralysis	124	1	194	1.2	0.0538
Other neurological disorders	767	6.4	1096	7.2	0.0044
Chronic Obpulmonary disease	1182	9.8	1828	12	<0.0001
Diabetes, uncomplicated	1244	10.4	1569	10.4	0.9008
Diabetes, complicated	721	6	1031	6.8	0.0056
Hypothyroidism	432	3.6	477	3.2	0.0489
Renal failure	772	6.4	942	6.2	0.5427
Liver disease	478	4	703	4.6	0.0063
Peptic ulcer disease, excluding bleeding	86	0.8	149	1	0.0165
AIDS/HIV	117	1	160	1	0.4804
Lymphoma	63	0.6	56	0.4	0.0582
Metastatic cancer	172	1.4	227	1.4	0.6195
Solid tumor without metastasis	589	4.8	691	4.6	0.2147
Rheumatoid arthritis/collagen vascular diseases	326	2.8	355	2.4	0.0599
Coagulopathy	290	2.4	417	2.8	0.0722
Obesity	1800	15	2085	13.8	0.0068
Weight loss	484	4	571	3.8	0.3035
Fluid and electrolyte disorders	1423	11.8	2128	14	<0.0001
Blood loss anemia	38	0.4	44	0.2	0.714
Deficiency anemia	404	3.4	565	3.8	0.0919
Alcohol abuse	265	2.2	495	3.2	<0.0001
Drug abuse	604	5	1140	7.6	<0.0001
Psychoses (mental illness)	243	2	497	3.2	<0.0001
Depression	1133	9.4	1564	10.4	0.0105

## Data Availability

Not applicable.

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
