# Peer review of "Disparities in Healthcare Utilization: Superfund Site vs. Neighboring Comparison Site"

_ijerph, 2022, doi:10.3390/ijerph19159271_

Round 1
Reviewer 1 Report
This is an interesting paper that examines healthcare utilization among patients living in zip codes within Jefferson County Alabama that may have been impacted by Superfund hazardous waste sites. It also characterizes differences in population health between these areas with a neighboring control area. Overall, this paper uses a powerful source of EHR data to provide important public health insights. Suggestions for improving the manuscript are included below.
- Line 59: cite the source for the data analysis.
- A citation is needed for line 104-107. Provide examples from these prior study findings.
- I suggest deleting or condensing lines 114-121 since detailed geographical results for DC are not relevant for Jefferson County. Also, is there any other relevant research for other cities or areas within the coal mining belt with similar issues?
- line 120: a space is needed after PM2.5. Also, do you mean “sources” instead of “resources”?
- line 126: Suggest rephrasing to “This burden on population health…” instead of “The burdens”.
- line 129-30: Suggest expanding to: “is critical for timely and effective public health intervention”.
- line 130-1: This sounds too editorial. Suggest deleting or re-writing this sentence.
- line 132-140: I’m not sure what the relevance of this paragraph is to the study, which utilizes quantitative analysis of EHR data and does not involve stakeholder interviews. Suggest deleting this section.
- line 141-144: either delete or incorporate into the Discussion.
- Overall, the Methods section is brief and lacks detail, especially on the chosen measures.
- Line 152: What study area? Clarify that this 64% relates to the total population in zip codes 35207, 35217 and 35214 (if this is correct). What is this percentage across the three sites? Also, please provide details on how these zip codes were selected. Add a sentence on how ‘affected’ is defined, ie, location of superfund. Were these the only two zip codes with superfund sites? Add a sentence on how the control zip code was selected. Lines 155-160 should be in the results or discussion sections.
- line 154 and elsewhere: data is plural.
- 164: “or” should be deleted before “diagnosis codes”.
- Table 1: add column headings for number and percent. Consider using “Socio-demographics” or “Socio-economic factors” instead of “Demographics” since you have education and financial concern measures. In Methods, explain how financial concerns are measured.
- line 195 and Table 2: Why weren’t the other characteristics in Table 1 adjusted for? For example, the differences in utilization may relate to the differences in smoking and substance use as well as other factors.
- Table 3 should be before Table 2 to be consistent with the text.
- In Methods and Table 2, clarify that the utilization outcomes were dichotomous. Change “Estimated” to “Adjusted”. “Logistic regression-adjusted odds of any healthcare utilization by setting”.
- Table 3: Reword to “Number of patients with any Inpatient visits”. Same for other two settings. The column headings are miss-spaced. Also, CI’s are preferred to SD’s.
- Line 196: Where are these adjusted results? Table 4 is unadjusted, right?
- Table 4: Here and elsewhere, please use more descriptive titles for exhibits, eg, “Prevalence of Elixhauser comorbidity conditions stratified by location”. Where is “Lower respiratory tract infections”? Should “Chronic pulmonary disease” be “chronic obstructive pulmonary disease”? Also, provide a list of the ICD-10 codes used in an appendix.
In discussion, generally discuss how some of the highest prevalent physical and mental health conditions can be directly or indirectly associated with living near a superfund site. Eg, aside from the direct environmental insults, lower SES individuals are more likely to locate in these areas due to financial constraints. Lower SES is associated with poor access to care and poor health status. The direct environmental insults compound and aggravate this situation with SES and health status.
- line 205: Incorporate 3.3 with 3.1.
- Line 223: I’m not sure that congestive heart failure should be included as a "pulmonary disease". Also,“lung” may be redundant with “pulmonary”.
- 228: add citation on this. How much higher are the air pollution levels? “affected areas”.
- 234: change the semi-colon to a comma.
- 245: This should read “had 14% lower adjusted odds of having an OP visit…”. A limitation to the study is lack of data on office-based visits.
- 251: “Our data analysis reveals…”. Also, in lines 251-4, do not restate specific results in reference to Table 4 in Discussion. This should go into Results.
- 255-6 is awkwardly phrased. Re-write as a separate sentence. Add citation for line 256-8. See my comment above regarding Table 4 for discussion.
- 260: what is a “50-pack-year” history? Re-word to “history of chain smoking”.
- 270: avoid loaded terms such as “bleak”. Suggest “substantial adverse health outcomes” or “substantial negative impacts on population health”.
- 271-3: This sentence seems odd starting with “one mission…”. Re-word to re-state or emphasize the study’s goal or objective. In general, avoid editorial, advocacy or superfluous language in research papers.
- Line 274 paragraph: this should be in Discussion. Conclusions should be concise and summarize the major takeaways and public health implications from the study. “largely unchanged” where? In Jefferson County?
- 281 paragraph: move to Discussion. 286: This paper relates to a domestic issue; discuss how the US can improve its own legal mechanisms to address the study findings. Line 287: briefly discuss specific future studies that are needed to examine or address this issue in JC based on the study findings.
- 290 paragraph should go into Discussion.
- 293: the matching process and determination of ‘equivalent data availability” need to be described in Methods.
- 300: “to if” should be “of whether”
- 302-4: “based on prior literature.”
- 304: delete “to” before “how factors”. Also, add “these” in front of “factors”.
- 305: “geographic” areas
- 308: “dissimilarities in health status between our affected and control areas”
- Another limitation is lack of data on migration and length of residency and thus length of exposure to environmental insults in the zip codes.
Author Response
Authors thanks to reviewer 1 for thought comments and suggestions. We addressed all the comments/edits.

Reviewer 2 Report
This is a very interesting manuscript and I think will be a great publication upon revision. While I think most of the necessary elements are there, they can be reorganized/improved on as follows:
- My biggest critique of the manuscript is the overall objective is vague; I'm not sure the overall exposure-response relationship(s) for which the authors want to provide evidence. Is the main exposure the sociodemographic variables or pollution (using area code as a proxy)? As such, it seems like multiple analyses strung together without an overarching scientific narrative. Inclusion of an epidemiological model eg a DAG would clarify this for the reader and help the authors direct the analyses in this regard, especially in terms of covariates included in the model(s).
- It would be good to note the major findings of the study in the abstract
- Section 1.1: it would be beneficial to give more information as to the selection of the different zip codes, including the comparison. Perhaps a map would help. It would also lead to a better understanding of potential distribution of pollution from the Super Fund site
- Section 1.2: This section is unclear to me
- The statistical analysis needs more detail
- The authors benefit from a large sample size which allows precise estimates and high statistical power. However, because of this statistical significance is common even with small differences (eg 1-2%). As such, it would be worthwhile to include a discussion of the clinical relevance of some differences.
- L176-179: This fits better after the unadjusted results are presented in the following paragraph
- Table 2: I'm not sure these are the only potential confounders and, as such, the epidemiological model suggested above would help in variable selection. Additionally, I believe this table would do better following Table 3 as it would then present the adjusted analysis after the adjusted.
- Table 3: I would love more discussion of the last 4 rows of this table
- Section 3.3: This discussion and Table 2 most suffer from the lack of clarity as to the intended objective for this study. Both pollution (which can lead to disease and exacerbate disease) and these variables can effect the rate of healthcare usage. Treating the analyses separately suffers from the potential of confounding and, thus, blurs the contribution of each independently
- Discussion: the authors use "rate" when they mean odds and given the high proportion of the outcome (eg OP visits) an OR is not a good estimate for RR. The authors also note the potential sample bias in their analysis however I think they should emphasize more it most as those with missing data could be the most disenfranchised and an important subset of the population.
- Conclusion: strengths and limitations of the study I believe are more appropriate for the discussion.
Author Response
Thank you very much for thought comments and edits. We addressed reviewer's comments.
